# Innovative Insights into Single-Cell Technologies and Multi-Omics Integration in Livestock and Poultry

**DOI:** 10.3390/ijms252312940

**Published:** 2024-12-02

**Authors:** Ying Lu, Mengfei Li, Zhendong Gao, Hongming Ma, Yuqing Chong, Jieyun Hong, Jiao Wu, Dongwang Wu, Dongmei Xi, Weidong Deng

**Affiliations:** 1Yunnan Provincial Key Laboratory of Animal Nutrition and Feed, Faculty of Animal Science and Technology, Yunnan Agricultural University, Kunming 650201, China; yinglu_1998@163.com (Y.L.); mfli_2000@163.com (M.L.); zander_gao@163.com (Z.G.); mhm1283403564@163.com (H.M.); 2022004@ynau.edu.cn (Y.C.); hongjieyun@163.com (J.H.); 15229238680@163.com (J.W.); danwey@163.com (D.W.); 2State Key Laboratory for Conservation and Utilization of Bio-Resource in Yunnan, Kunming 650201, China

**Keywords:** single-cell RNA sequencing, multi-omics integration, cellular heterogeneity, transcriptional regulation, livestock and poultry biology

## Abstract

In recent years, single-cell RNA sequencing (scRNA-seq) has marked significant strides in livestock and poultry research, especially when integrated with multi-omics approaches. These advancements provide a nuanced view into complex regulatory networks and cellular dynamics. This review outlines the application of scRNA-seq in key species, including poultry, swine, and ruminants, with a focus on outcomes related to cellular heterogeneity, developmental biology, and reproductive mechanisms. We emphasize the synergistic power of combining scRNA-seq with epigenomic, proteomic, and spatial transcriptomic data, enhancing molecular breeding precision, optimizing health management strategies, and refining production traits in livestock and poultry. The integration of these technologies offers a multidimensional approach that not only broadens the scope of data analysis but also provides actionable insights for improving animal health and productivity.

## 1. Introduction

Cells are the basic functional units of living organisms, laying the foundation for life formation and evolution through complex biochemical reactions and self-replication abilities. Since Robert Hooke’s discovery of cells in the mid-17th century [1], cell biology has undergone centuries of progress—from optical microscopy to fluorescence microscopy [2,3], from flow cytometry [4,5] to multi-sensor cell-sorting instruments [6], and finally modern scRNA-seq technologies. In recent years, high-throughput multi-omics technologies including genomics [7], epigenomics [8], transcriptomics [9], proteomics [10], and metabolomics [11] have been increasingly used in livestock and poultry research. However, traditional bulk-level analysis methods, such as bulk RNA-seq, though capable of revealing gene expression patterns in intact tissue samples, fail to discern differences between cell types, especially in tissues with significant cellular heterogeneity [12]. To address this limitation, scRNA-seq emerged, providing single-cell resolution to reveal information about gene expression patterns, enabling a deeper understanding of cellular heterogeneity and regulatory biological processes (Table 1).

scRNA-seq provides new tools for studying complex regulatory mechanisms in cell development, immune responses, and reproduction. In poultry and livestock research, single-cell transcriptomics technologies have shown great potential, especially in profiling the immune system, developmental processes, and disease responses [13]. For example, scRNA-seq has helped to reveal the specific roles of different cell types in the immune response or to reveal the transcriptomic profile of key cells during reproduction [14]. However, single-cell analysis in livestock and poultry research still faces challenges such as cost, technical complexity, and data integrity, which limit its dissemination in practical applications [15]. Moreover, gene expression data captured by this technique are often limited to the RNA level, making it difficult to directly reflect other biological information, such as proteomics and metabolomics [16]. Consequently, there is a growing trend towards integrating scRNA-seq with other omics layers—such as epigenomics, proteomics, and spatial transcriptomics—yielding a multidimensional analytical approach. This integrated perspective not only deepens our understanding of gene regulatory networks but also supports a comprehensive analysis of cellular states and biological processes.

This review will comprehensively discuss the current application status of scRNA-seq in livestock and poultry research, explore recent advances in single-cell and multi-omics integration, and assess its potential applications in molecular breeding, production performance optimization, and disease prevention, with the aim of providing novel insights and methods for livestock and poultry research.

## 2. Fundamental Principles of scRNA-seq

Despite the fact that all cells, tissues, and organs within a single organism share an identical genome [17], different cell types exhibit unique RNA expression profiles. These variations lead to diverse protein synthesis and cell functions, enabling cells with the same genome to perform a variety of biological tasks, thereby contributing to biological diversity and complexity [18]. Traditional RNA-seq methods analyze overall transcriptomic information by extracting total RNA from tissue samples, yet they are unable to distinguish the specific expression patterns of different cell types. This limitation hampers their application in heterogeneous tissues [19]. To overcome this challenge, scRNA-seq was first applied by Tang et al. in 2009 to a four-cell-stage mouse embryo, revealing the transcriptomic characteristics of individual cells [20]. This technology has revolutionized our ability to dissect cellular heterogeneity. By 2013, scRNA-seq was recognized as one of the most promising research areas by *Science* and named “Method of the Year” by *Nature Methods*, underscoring its importance in life sciences. With the launch of the international “Human Cell Atlas” project [21], efforts to map cell types in humans and non-model organisms have been greatly accelerated.

In recent years, scRNA-seq technology has increasingly been applied in the study of economically valuable livestock and poultry, such as pigs [22], cattle [23], sheep [24], chickens [25], and ducks [26]. These species are not only vital in agricultural production but also serve as ideal models for investigating complex processes in developmental biology [27], cell differentiation, and reproductive biology [28]. The widespread application of scRNA-seq not only advances foundational biological research in these animals but also paves the way for practical applications in molecular breeding and disease prevention.

The first step in scRNA-seq is the isolation of target cells from a sample, a critical process that requires maintaining cell viability and isolating individual cells from a heterogeneous cell population [29]. Common cell isolation techniques include serial dilution [30], fluorescence-activated cell sorting (FACS) [31], micromanipulation [32], microfluidics [33], immunomagnetic separation [34], and laser capture microdissection (LCM) [35]. Each method has distinct characteristics, varying in their efficiency, speed, and impact on cell integrity (Table 1). The sample preparation and sorting conditions significantly affect the efficiency of single-cell isolation and cell viability. Thus, improving separation techniques and developing alternative methods [36] are key directions for enhancing scRNA-seq data quality.

After cell isolation, the next essential step is library preparation and sequencing. The core of library construction involves the reverse transcription of RNA molecules into cDNA followed by amplification. Key factors for library quality include sample integrity, appropriate library concentration, fragmentation degree, end repair, and adapter ligation accuracy [37]. Each step requires precise control to maximize library quality and, consequently, the accuracy and biological significance of the sequencing data. Current scRNA-seq amplification methods primarily use PCR or in vitro transcription linear amplification [38]. PCR-based methods are further classified into scRNA-seq, Smart-Seq, Stochastic Reversible Termination Sequencing (STRT-Seq), Cell-Seq [39], and Polymerase-Mediated Amplification Sequencing (PMA-Seq) [40] (Figure 1 and Table 2). Studies show that the total RNA in mammalian cells averages around 10 pg, with an mRNA abundance as low as 0.1 pg [41]. Thus, scRNA-seq requires highly efficient whole-transcriptome amplification techniques to capture as comprehensive a gene expression profile as possible. Whole-transcriptome sequencing not only provides abundant transcriptomic data but also lays the groundwork for a detailed analysis of cellular biological properties.

Following library construction, data processing includes quality control, sequence alignment, expression quantification and normalization, dimensionality reduction, clustering, and differential gene expression analysis [42]. Among scRNA-seq platforms, the 10× Genomics system is one of the most widely used. It is particularly good at high-throughput single-cell transcriptomics research using microfluidics [43]. This platform leverages microfluidics to combine a lysed single-cell suspension with barcoded gel beads in oil droplets, enabling high-throughput cell capture [44] and 3′ transcriptomic (mRNA) gene expression profiling [45]. This method allows for the parallel sequencing of numerous single cells and enables the precise identification of cell types of interest in subsequent analyses. The development of scRNA-seq technology has shifted transcriptomics research from a bulk to a single-cell level, achieving a transition from qualitative to quantitative analysis. These advancements offer critical tools for deciphering gene expression regulation, driving a deeper understanding of complex biological processes.

**Table 2 ijms-25-12940-t002:** Comparison of scRNA-seq library preparation methods.

Methods	Coverage Area	Read Depth	UMI	Amplification Method	Strand-Specific	Characteristics	Reference
Tang 2009	Full length	10^4^–10^5^	No	Homopolymer tailing	No	High sensitivity and comprehensive transcriptome coverage	[46]
Strt-seq	5′ end	10^4^–10^5^	Yes	Template switching	Yes	Captures complete transcript data, suitable for gene fusion studies	[47]
Smart-seq	Full length	10^6^	No	Homopolymer tailing	No	High sensitivity, captures low-abundance transcripts	[48]
Drop-seq	3′ end	10^4^–10^5^	Yes	In vitro transcription	Yes	High throughput, cost-effective for large sample sizes	[49]
Cel-seq	3′ end	10^4^–10^5^	Yes	In vitro transcription	Yes	High sensitivity, low bias in expression quantification	[39]
Mars-seq	3′ end	10^4^–10^5^	Yes	In vitro transcription	Yes	High throughput with robust data coverage	[50]
Cyto-seq	3′ end	10^3^–10^4^	Yes	Designed primers	No	Ideal for cell surface labeling and functional studies	[51]
10× Genomics	3′ end	10^4^–10^5^	Yes	Template switching	Yes	Sensitive detection of low-abundance genes in heterogeneous cells	[43]
Scrb-seq	3′ end	10^4^–10^5^	Yes	In vitro transcription	Yes	Suitable for studying cellular heterogeneity and dynamics	[52]

## 3. Applications of scRNA-seq in Livestock and Poultry

scRNA-seq has successfully mapped cell atlases in various model organisms, such as mouse embryos [53], Drosophila brains [54], and zebrafish telencephalons [55]. However, its application in economically important livestock and poultry, focusing on growth, development, and trait formation, remains limited. Below, we highlight recent advances in scRNA-seq research in livestock and poultry and its biological implications.

### 3.1. Applications of scRNA-seq in Poultry

Poultry is a major global food source, providing high-quality protein and playing a vital role in the agricultural economy [56]. With the increasing demand for quality protein, enhancing poultry production performance, breeding efficiency, and health has become a research focus [57]. By elucidating regulatory mechanisms at the gene, cell, and molecular levels, scRNA-seq offers novel perspectives and tools for poultry genetic breeding and production optimization.

One of the major strengths of scRNA-seq lies in its capacity to resolve cellular heterogeneity, which is particularly advantageous in studying poultry gonadal development. Support cell lineages in the gonads are among the earliest differentiating cell types in development and are crucial to gonadal growth, function, and regulation [58]. Nevertheless, gonadal differentiation shows significant interspecies differences [59]. For instance, scRNA-seq analysis comparing gonadal data from Hy-Line Brown chicken embryos and mouse embryos revealed that chicken support cell lineages do not originate from the coelomic epithelium, but rather from a mesenchymal cell population expressing DMRT1+/OSR1+/PAX2+/WNT4+. Additionally, *Pax2* gene migration from the kidney to the gonads may serve as an early marker for chicken support cell progenitors [25]. Beyond gonadal development, scRNA-seq demonstrates powerful capabilities in exploring other developmental systems. By analyzing chicken embryos at Hamburger–Hamilton (HH) stages 25, 29, and 31 using scRNA-seq, the transcriptional dynamics during development were profiled at single-cell resolution, identifying 23 cell clusters and highlighting *RSPO3* as a key marker of the chicken apical ectodermal ridge (AER) [60]. Another study on avian limb skeletal development identified Fox/Sox binding sites in the Gdf5-associated regulatory region (GARR) as influential for joint surface structure and Lmx1b binding sites in the GARR as regulators of elbow joint symmetry along the dorsoventral axis [61]. These findings underscore the critical applications of scRNA-seq in avian limb and skeletal development research.

scRNA-seq also holds substantial promise for economic trait research. Analysis of the pectoral muscle in chickens at days 5 (D5) and 100 (D100) unveiled skeletal muscle heterogeneity, revealing different cellular clustering patterns across developmental stages. The D5 stage showed an abundance of myogenic and adipocyte clusters, reflecting incomplete myogenic differentiation. RNA in situ hybridization validated *APOA1* and *COL1A1* as marker genes for intramuscular fat [62]. During egg production trait studies, the scRNA-seq profiling of Shaoxing duck livers at different laying stages identified VLDL II (APOV1) and APOB as activated during egg production, while APOA1, A4, and C3 were found to contribute to cessation [26]. These findings provide critical insights for identifying the biomarkers of increased-egg-laying traits and selective breeding in poultry. In neural development, scRNA-seq similarly shows unique advantages. Analyzing the evolution of the avian embryonic ectoderm from the primitive streak to the neuralization stage, scRNA-seq revealed cellular diversity and heterogeneity in the ectoderm, particularly demonstrating that neural plate border (NPB) cell fates emerge from the dynamic expression of competing transcriptional programs rather than discrete transcriptional states. The same study proposed a probabilistic model of cell fate determination, emphasizing the spatiotemporal significance of progenitor cells and their progeny in fate decisions [63].

In summary, the extensive application of scRNA-seq in poultry research has not only revealed complex regulatory mechanisms underlying cellular heterogeneity and developmental processes but has also provided new perspectives on trait formation, muscle development, and reproductive regulation. These findings offer critical references and support for improving poultry production performance, optimizing breeding strategies, and implementing informed management practices.

### 3.2. Application of scRNA-seq in Pigs

As one of the most widely consumed meats worldwide, pork is an essential protein source due to its high yield and efficient feed conversion. With rising demand for pork production and quality in the livestock industry [64], scRNA-seq offers valuable insights into pig growth, immune function, and reproductive biology, providing opportunities to enhance production efficiency and refine livestock management.

A study applying scRNA-seq focused on the skeletal muscle cell populations of newborn pigs of different breeds, identifying nine cell clusters. The findings indicated a significant increase in fibro-adipogenic progenitors (FAPs) and a decrease in muscle cells in domestic pigs compared to wild boars. Among the breeds, Duroc pigs had the highest proportion of endothelial cells, myeloid cells, and proliferative pre-adipocytes, while Laiwu pigs had the highest glial cell content, suggesting lean and obese pig breeds have varying myogenic potentials likely due to selective breeding for muscle growth efficiency. Additionally, the study observed a significantly lower proliferation of myogenic progenitor cells in wild boars, implying domestic pigs have greater muscle growth potential [65]. Further research demonstrated that FAP cells secrete factors such as FGF, TGFb, and WNT that interact with relevant receptors on myogenic cells to support muscle development. FGF7 and FGFR2 are highly expressed in pig muscle satellite cells (MuSCs), with FGF7 levels influencing MuSC proliferation. Knocking out FGFR2 significantly reduced the proliferative effect, highlighting the FGF7-FGFR2-mediated interaction between FAPs and MuSCs, promoting muscle regeneration and repair [66].

In reproductive research, scRNA-seq based on 10× Genomics and Smart-seq2 identified distinct subpopulations of cumulus granulosa cells (cGCs) and mural granulosa cells (mGCs) within porcine antral follicles, revealing their differential roles in estrogen synthesis and hormone signaling. Specifically, the mGC2 subpopulation was involved in estrogen synthesis, while mGC1 regulated hormone signaling, and cGC2 primarily contributed to glycolysis and cumulus expansion, providing insights into the mechanisms of follicular development and oocyte maturation in pigs [67]. Regarding the immune system, an scRNA-seq analysis of peripheral immune cells in healthy Large White pigs identified 14 cell types. Within the αβ T cell lineage, CD8 T cells differentiated from CD8 naïve to CD8 memory states. A GO analysis showed CD8 naive cells were enriched in ribosome synthesis-related functions, while memory and effector CD8 T cells primarily participated in immune response processes, highlighting functional differences and high peripheral blood cell heterogeneity [68]. Another study examined differential gene expression in pig thoracolumbar and rib development, finding that the *HOXA10* gene was nearly absent in thoracic cells but highly expressed in lumbar osteoblasts, with localization in a non-coding region. This suggests that *HOXA10* may play a regulatory role in the thoracolumbar transition, offering new perspectives on skeletal development mechanisms [69]. By integrating scRNA-seq and snRNA-seq data, a single-cell transcriptomic atlas of 20 porcine tissues was constructed, identifying 234 distinct cell clusters and 58 main cell types. Vascular endothelial cells (VECs) displayed substantial heterogeneity across tissues, with 21 VEC subtypes showing unique expression profiles and functions, including proliferative and immunoactive endothelial cells. CellChat analysis and experimental validation identified a mesenchymal transition cell subtype in adipose tissue and revealed that the TGF-β2 signaling pathway plays a crucial role in this process [70].

In brief, scRNA-seq applications in various pig tissues and biological processes have revealed cell population heterogeneity and complex molecular regulatory mechanisms, deepening our understanding of porcine skeletal muscle growth, follicular development, vertebral formation, and immune cell function. These studies offer valuable theoretical insights for enhancing pork production efficiency and optimizing livestock management.

### 3.3. Application of scRNA-seq in Ruminants

With advancements in single-cell transcriptome sequencing in ruminants, more research has focused on dynamic gene expression changes during spermatogenesis and oocyte development [71]. For instance, the scRNA-seq profiling of bovine oocytes has elucidated growth-associated genes and gene clusters, identifying a positive correlation between oocyte growth and key genes (*DPPA3*, *ESRRG*, and *WEE2*) and a downregulation trend in oxidative phosphorylation-related genes [72]. Additionally, research on Hu sheep ovaries revealed significant gene expression differences in granulosa cells (GCs) between primiparous and multiparous ewes, with the high expression of *JPH1* promoting GC maturation, while LOC101112291 was downregulated during early GC differentiation into cumulus cells. Mature GCs expressed *FTH1* and *FTL*, which modulate intracellular iron levels and oxidative stress to facilitate oocyte development [73].

In male reproduction, scRNA-seq generated a high-resolution transcriptomic atlas of buffalo testes, identifying two spermatogonial stem cell clusters (SSC-1 and SSC-2), expressing *NANOS2* and *RBM47*, respectively. These data reveal a significant histone-to-protamine transition during spermatogenesis, marking the gradual maturation of sperm [74]. Similar findings in yak testes identified SSC clusters expressing *BMX* and *PTGS2*, with early spermatocytes expressing *SYCP3*, further elucidating the molecular mechanisms involved in spermatogenesis [75]. Studies on Guanzhong dairy goats revealed that highly expressed differential genes such as *EZH2*, *SCP2*, *PRKCD*, and *PCNA* during spermatogenesis were mainly enriched in the Notch, Hippo, TGF-β, and stem cell pluripotency pathways, identifying *AES* and *TKTL1* as specific marker genes for spermatogonia in dairy goats [76]. These findings offer critical molecular insights into oocyte and sperm development in ruminants. Single-cell quantitative microfluidics were also used to identify early developmental genes in bovine embryos and revealed that NOTCH1 is involved in early cell fate decision and tissue differentiation, and that the late-stage-specific expression of *TBX3* and *FGFR4* is associated with cell proliferation and migration. Further, the primitive endoderms and epiblasts of bovine blastocysts showed that the specific expression of *TDGF1* and *PRDM14* may be involved in early cell fate decisions [77].

In somatic cell nuclear transfer (SCNT) research, scRNA-seq has also shed light on the developmental mechanisms of SCNT embryos. The abnormal expression of genes such as *POLR2K*, *GRO1*, and *ANKRD1* at the eight-cell stage and the lack of ZSCAN4 activation impair the transition from fertilization to the morula stage in SCNT embryos [78]. Research on Holstein bull calves further used scRNA-seq to analyze satellite cell clusters isolated from the longissimus dorsi (LD) muscle at 14 days of age, identifying 15 clusters, including myoblast subsets and satellite cell subpopulations with varying myogenic states. The study also traced the origin of FAPs in skeletal muscle [79]. Research into ruminant growth performance will directly impact meat quality and nutritional value, so in-depth scRNA-seq investigations into myoblast development and differentiation reveal molecular mechanisms underlying growth performance.

To sum up, the application of scRNA-seq in livestock research offers fresh insights into growth, development, and reproductive processes. First, it reveals dynamic cellular changes across developmental stages, facilitating the analysis of gene expression profiles in specific cell populations and temporal transitions. Second, identifying genes associated with economic traits, such as growth rate, meat quality, and egg production, provides valuable regulatory information to enhance productivity and breeding strategies. Finally, this technology provides high-resolution data supporting molecular breeding and genetic improvement, advancing precision breeding.

## 4. Application of Multi-Omics Integration with scRNA-seq in Livestock and Poultry

While scRNA-seq technology enables gene expression analysis at the single-cell level, it predominantly captures gene activity at the RNA stage [80]. Despite advances in single-cell genomics, epigenomics, proteomics, and metabolomics techniques, challenges such as low cell throughput, species-specific limitations, high costs, and development barriers hinder comprehensive genetic mapping [81]. For livestock, complex physiological characteristics and diverse production environments compound these limitations [82]. Thus, integrating traditional genomics or multi-omics approaches with scRNA-seq offers an optimal strategy for unraveling complex traits in livestock, such as growth, immunity, and disease resistance. This combined approach can provide valuable insights for enhancing animal health, productivity, and genetic selection programs.

### 4.1. Integrated Analysis of Single-Cell Transcriptomics and Epigenomics

Epigenomics sequencing is used to explore heritable changes in gene expression and regulation, encompassing 3D chromatin structure, chromatin accessibility, histone modifications, DNA methylation, and RNA methylation. These analyses can be divided into the epigenome [83] and epitranscriptome layers [84]. Common epigenomics sequencing techniques include Assay for Transposase-Accessible Chromatin with high-throughput sequencing (ATAC-seq) and Chromatin Immunoprecipitation Sequencing (ChIP-seq), both of which are widely applied in livestock research [85].

ATAC-seq, developed in 2013 by Dr. William Greenleaf, is a high-throughput sequencing method that profiles the accessible regions of chromatin by inserting NextEra adaptor sequences into open chromatin regions using a modified Tn5 transposase (Figure 2A) [86,87]. The transposase also cleaves the DNA via its transposition activity, thereby enabling the construction of next-generation sequencing (NGS) libraries [88]. This technique effectively maps chromatin accessibility, identifying potential enhancers, repressors, and key transcription factors (TFs). In recent years, ATAC-seq has often been combined with scRNA-seq to integrate gene expression with chromatin accessibility in key cell populations, enabling the more accurate identification of gene regulatory elements. For example, a combined scRNA-seq and scATAC-seq study of cattle LD tissue development revealed distinct cell types across the fetal, postnatal, and adult stages, and identified core TFs like MSC, MYF5, and GLI1 involved in directing cellular fate changes in skeletal myogenesis [89]. In chickens, snRNA-seq and snATAC-seq have been applied to gonadal tissue analysis at different developmental stages, constructing maps of gene expression and chromatin accessibility at the single-cell level. This has led to the further elucidation of sex-determining genes, specific cis-regulatory elements and TFs, highlighting TFs such as DMRT1 and ESRRG, which potentially regulate the dual-potential fate of supporting cells in sex determination. Comparative analyses across species have shown high conservation of TFs, such as DMRT1, TOX3, and WT1 [90]. Collectively, integrated analysis using ATAC-seq and scRNA-seq offers several advantages: First, the dual data on gene expression and chromatin accessibility enhance cell-type identification and classification precision. Second, it provides insights into regulatory mechanisms of specific genes and their roles across cell types or developmental stages. Lastly, assessing chromatin structure changes allows an exploration of the underlying factors driving cellular functional properties and fate decisions.

ChIP-seq is a genome-wide technique used to map specific histone modifications (e.g., H3K4me1, H3K4me3, H3K27ac) or DNA modifications and TF binding locations across the genome [91]. This is achieved by immunoprecipitating chromatin–protein complexes followed by high-throughput sequencing [92]. Although ChIP-seq provides valuable insights into transcriptional networks, it faces challenges, such as requiring high-quality cells and preventing nonspecific binding [93]. Integrating ChIP-seq with scRNA-seq enables a more accurate interpretation of the transcriptional regulatory roles of distinct cell types, helping to construct comprehensive gene regulatory networks. Combined bulk RNA-seq and scRNA-seq analyses in humans and cattle revealed high expression of Androglobin (ADGB) in the female reproductive tract, with ADGB shown to be co-regulated with FOXJ1 in cilia formation. ChIP-seq results further revealed FOXJ1 binding to the ADGB promoter region, indicating its role in the transcriptional regulation of ADGB through synergistic interactions with RFX2 to stabilize the activation of ciliary genes [94]. In studies examining follicular development in chickens, integrating bulk RNA-seq, scRNA-seq, ChIP-seq, ATAC-seq, and Hi-C data has revealed characteristic histone H3K27ac marks associated with active enhancers in granulosa cells (GCs) during follicular development. This integrative approach has shown that gene expression and chromatin structural changes are concordant, underscoring how transcriptional and chromatin GC changes foster transcriptional activity during follicle development [95]. The integration of ChIP-seq with other genomic data advances our understanding of transcriptional regulation and reveals how chromatin structure dynamics influence gene expression in developmental, immune, and other biological processes.

In sum, the integration of ATAC-seq and ChIP-seq with scRNA-seq provides powerful tools for studying chromatin structure, epigenetic modifications, and gene regulatory mechanisms. With the continuous advancement of new technologies, such as CUT&Tag, the integration of epigenomics and multi-omics will open up broader research opportunities in livestock studies, providing a comprehensive and multidimensional map of gene regulation.

### 4.2. Integrative Analysis of Single-Cell and Bulk RNA-seq

Bulk RNA-seq and scRNA-seq are two key branches of transcriptomics, each offering different scales and resolutions of analysis [96]. scRNA-seq reveals cellular heterogeneity by providing gene expression information at the single-cell level, but it is often limited by high costs and technical demands, making it challenging to capture a comprehensive gene expression profile [97]. Conversely, bulk RNA-seq offers a cost-effective means of providing a more complete gene expression profile but cannot resolve cellular heterogeneity, potentially masking cell-specific information due to averaging effects [18]. For example, a study of spermatogenesis in sheep using scRNA-seq identified early and late primary spermatocytes, round spermatids, and other germ cells [24]. In contrast, combining these two approaches to study spermatogenesis in Guanzhong pigs resulted in the identification of not only spermatogonia, spermatogonia, and spermatocytes in the testes, but also three new types of somatic cells, in addition to the validation of germ cell types using bulk RNA-seq data and the discovery of new key markers for the porcine spermatogonial subpopulations *CD99* and *PODXL2* [98]. These findings indicate that combining these approaches can more fully elucidate the molecular mechanisms underlying germ cell development.

In an integrated analysis of epithelial fold development at the uterovaginal junction (UVJ) of White Leghorn chickens, bulk RNA-seq revealed that genes such as *BMP4*, *FGF10*, *SOX10*, and *EDN2*, as well as the WNT and TGF-β signaling pathways, regulate cell proliferation, differentiation, and migration. scRNA-seq identified 17 cell clusters, with carbonic anhydrase playing a key role in UVJ epithelial fold formation; CA4 directly regulated the epithelial cell (EpiC) acid–base balance and bicarbonate transport, while CA2- and CA8-expressing cells indirectly promoted development through intercellular signaling and metabolic regulation [99]. Additionally, an scRNA-seq and bulk RNA-seq analysis of a porcine heart failure model, treated with antimiR-21, identified macrophages and fibroblasts as key cell types. Deconvolution analysis indicated that antimiR-21 suppressed miR-21 expression, relieving the repression of its target genes and thereby reducing inflammation and MAPK signaling pathway activity [100]. In studies of intervertebral disk (IVD) degeneration, combined scRNA-seq and bulk RNA-seq analyses identified cell clusters in bovine caudal IVD nucleus pulposus (NP) and annulus fibrosus (AF) tissue, revealing notochord-associated markers like CD24, LGALS3, and CDH2 in the NP, as well as multifunctional stem cell markers KRT15 and CD44, supporting the presence of notochord-like phenotypes in specific NP cell clusters. The AF cells predominantly expressed high levels of collagen and leucine-rich small proteoglycans like BGN and FMOD, suggesting that collagen-rich AF expression is essential in maintaining the structure, function, and biomechanics of the IVD [101]. In conclusion, the use of bulk RNA-seq to acquire overall gene expression data in conjunction with scRNA-seq, which provides finer cell-level analysis, collectively provide comprehensive insights that contribute to a deeper understanding of complex biological processes.

### 4.3. Integrated Analysis of Single-Cell Transcriptomics and Proteomics

As the primary executors of cellular functions, proteins are essential for maintaining cell structure, regulating metabolic reactions, and supporting overall cellular activity [102]. However, due to post-transcriptional regulation and modifications, mRNA abundance does not directly reflect protein abundance [103]. Recently, combining scRNA-seq with proteomics has gained traction, leveraging scRNA-seq data to identify differentially expressed proteins in proteomics analysis, facilitating cross-validation. This integrative approach provides deeper insights into the relationship between mRNA and protein levels, which enables a more precise characterization of cell types, cellular states, and dynamic changes under various developmental conditions.

A study comparing LD muscle in obese and lean pigs identified eight distinct cell types using scRNA-seq, focusing on the differentiation potential of Myo lineage cells. Its results indicated that lean pigs exhibited a higher proportion and differentiation potential of Myo lineage cells. Further proteomics analyses revealed that the differentially expressed proteins were enriched in pathways associated with muscle development, cellular proliferation, and differentiation. Additionally, lower cytoplasmic and endoplasmic reticulum Ca^2^⁺ levels in the muscle cells of lean pigs suggested that a reduced Ca^2^⁺ concentration might promote Myo lineage cell differentiation potential [22]. In another study involving a sinoatrial node model of the pig heart, the effects of GLP-1 receptor agonists were temporally mapped, with snRNA-seq showing that GLP-1 receptors were predominantly expressed in specific cardiomyocyte subtypes, overlapping with classical pacemaker cell markers. Combined proteomics analysis further indicated that GLP-1 receptor activation could modulate key proteins involved in sinoatrial node calcium signaling [104]. Emerging technologies are increasingly supporting single-cell proteomics, such as nanoPOTS, which is based on fluorescence-activated cell sorting [105]. For instance, a study using the nanoPOTS platform demonstrated that proteins such as OCM, CRABP1, and GPX2 were upregulated during hair cell differentiation in the chicken cochlea, while supporting cells exhibited a significant downregulation of proteins like thymosin β-4 (TMSB4X) and AGR3. Subsequent scRNA-seq further characterized gene expression patterns highlighted by proteomics, revealing that the downregulation of TMSB4X was crucial for proper hair cell differentiation. This process allowed these cells to utilize the increased availability of actin monomers to form stereocilia bundles, thereby supporting hair cell function and physiological activities [106].

On the whole, integrating and analyzing scRNA-seq with proteomics data can provide a comprehensive view of cellular-level information to more accurately resolve complex biological processes and regulatory mechanisms. By applying these combined techniques across species and tissues, researchers have not only uncovered differentiation potentials, regulatory pathways, and key molecular drivers of specific cell types but have also illuminated how factors like intracellular ion concentration and actin dynamics impact cellular functionality.

### 4.4. Integrated Analysis of Single-Cell Transcriptomics and Spatial Transcriptomics

Spatial transcriptomics (ST) was also named Method of the Year by *Nature Methods* in 2020, even though scRNA-seq was named Breakthrough of the Year [107]. While scRNA-seq enables the exploration of cell types and gene expression patterns over time, it lacks spatial specificity. The advent of ST fills this gap by quantifying the total mRNA in intact tissue sections, integrating spatial information with tissue morphology. This approach precisely quantifies and maps cellular distribution within tissues, creating complex gene expression landscapes [108]. Current ST methodologies are categorized into sequencing-based and imaging-based technologies. Sequencing-based ST, such as Visium HD [109] (Figure 2B), utilizes spatial barcoding and next-generation sequencing to provide a comprehensive view of gene expression patterns. In contrast, imaging-based ST applies fluorescence signals to RNA in situ and decodes them with high-resolution microscopy, ideal for studying specific protein or molecular spatial distributions, such as smFISH [110]. By effectively integrating scRNA-seq with ST, researchers can spatially resolve the dynamic changes in specific cell subtypes during developmental and physiological processes.

For instance, one study applied scRNA-seq and ST to embryonic pig skin to investigate the origin of early matrix cells, constructing a spatiotemporal transcriptional map. The results suggested that hair follicle matrix cells originated from an OGN+/UCHL1+ progenitor-like cell subgroup whose abnormal proliferation and migration contribute to hairless pig phenotypes [111]. Another study annotated eight myogenic cell subclusters in developing pig somites using scRNA-seq. Through pseudotime analysis, the researchers discovered that *EGR1* and its target gene *RHOB* were progressively upregulated as the proportion of progenitor cells decreased during skeletal muscle development, underscoring their critical role in myogenic differentiation. Further scATAC-seq data revealed that genes associated with myocytes and progenitor cells (e.g., *MSC*, *MYF5*, *MYOG*, and *PAX7*) exhibited high accessibility, with *EGR1* displaying significant motif enrichment in myocytes, establishing *EGR1* and *RHOB* as key regulators in porcine embryonic myogenesis [112]. Moreover, integrated scRNA-seq and ST analyses of developing chicken heart tissue (D4, D7, D10, and D14) revealed the upregulation of extracellular matrix (ECM) factors (*AGRN*, *EGFL7*, and *FN1*) involved in the epicardial-to-mesenchymal transition (EMT) essential for myocardial development. TMSB4X was notably enriched during coronary vascular development, where its knockdown led to cardiac developmental abnormalities [113]. During primordial follicle formation in chicken ovaries, scRNA-seq identified 24 cell types, while ST delineated 16 spatial clusters, revealing a differential expression of genes in pre-granulosa (e.g., *CYP19A1*, *HSD17B1*) and pre-theca cells (e.g., *NR5A1*, *DHCR7*) involved in steroid hormone biosynthesis, regulated by the estrogen receptor (ESR) pathway and its downstream effector GREB1 [114].

In conclusion, combining scRNA-seq with ST not only complements their respective strengths but also enables a comprehensive analysis of the gene expression and spatial relationships in cell populations, providing a novel perspective for understanding complex biological processes.

### 4.5. Integrated Analysis of Single-Cell Transcriptomics and Multi-Omics

Beyond ATAC/ChIP-seq, bulk RNA-seq, proteomics, and ST, other omics technologies—such as microbiomics [115] and metabolomics [116]—are increasingly integrated with scRNA-seq in livestock growth and development research. This integration enables a holistic analysis of molecular networks and system biology features in livestock. For instance, combining metabolomics with scRNA-seq dynamically captures intracellular metabolite changes, while microbiome integration reveals the impact of gut microbiota on immune and intestinal epithelial cells.

In the field of reproduction, a multi-omics analysis of Tibetan sheep provided key insights into the genetic mechanisms underlying fertility. Analyzing Tibetan sheep ovaries through GWAS, bulk RNA-seq, proteomics, and metabolomics, researchers found that PAPPA and the major reproductive gene *BMPR1B* were specifically expressed in mGC subpopulations. PAPPA promotes ovarian follicle growth and steroid synthesis, while mutations and the alternative splicing of *BMPR1B* affect the mechanisms of high fecundity in Tibetan sheep [117]. A high-resolution single-cell atlas of lactating dairy cows identified 55 major cell types, with metabolic characterization at a single-cell resolution revealing that Th17 cells are enriched in the foregut, interacting with a high short-chain fatty acid uptake EpiC subtype via IL-17 signaling to regulate nutrient transport [23]. Adaptation in livestock is a critical focus of animal research. By constructing a single-cell atlas of lung tissue in *Bos mutus* and *Bos grunniens* and comparing it with high-quality chromosome-level genomic data, researchers discovered that unique genes expressed in yak EndoCs, MSCs, and EpiCs exhibited high population genetic differentiation and structural variation (SV). Additionally, *Bos taurus* EndoCs showed a higher expression of *EPAS1* and *ADGRF5* compared to yak, suggesting a role in hypoxia adaptation, physiological regulation, and population-specific adaptation [118].

Advances in scRNA-seq have significantly enhanced our understanding of heterogeneity mechanisms in eukaryotic cell populations, and recent applications extend to prokaryotes, capturing bacterial functional heterogeneity in complex microbiomes [119]. Microbiome single-cell transcriptomics (Microbiome scRNA-seq), based on random priming and microfluidics, enables high-throughput RNA sequencing at the single-bacterium level [120]. Through microbiome scRNA-seq and pan-genome analysis, a single-cell transcriptomic atlas of the bovine rumen microbiome identified 12 functional cell clusters. HSP90 + HMAC clusters exhibited high activity in carbohydrate metabolism, and an Integrase + HMAC functional subcluster enriched in *Butyrivibrio* promoted pyruvate-to-succinate conversion in rumen carbohydrate metabolism [121]. Integrating 16S rDNA sequencing, metabolomics, and scRNA-seq to analyse the rumen revealed that adult cattle had fewer EpiC subtypes but significant metabolic functionality and redox-related activity. Additionally, a high co-occurrence of *Desulfovibrio* with pyridoxal was observed, collectively maintaining rumen redox balance [122]. In studies on metabolic disorders in dairy cows, the scRNA-seq of peripheral blood immune cells identified 10 immune cell types, with a significant downregulation of immune function in cows with excessive lipolysis [123]. Integrative analyses of the fecal 16S rRNA, metagenomics, and targeted metabolomics of blood and fecal bile acids indicated that excessive lipolysis in cows activated secondary bile acid biosynthesis, affecting *Bacteroides*, *Paraprevotella*, and *Treponema* in the gut. Comprehensive analysis revealed that reduced plasma glycodeoxycholic and taurocholic acids in lipolytic cows led to low GPBAR1 expression, promoting an immunosuppressive state in CD14+ monocytes [123].

In summary, combining scRNA-seq with multi-omics offers a novel perspective in animal science (Figure 3). Applications range from understanding the coordination of metabolic and immune functions during lactation in dairy cows to exploring the adaptive evolution of yaks and cattle, as well as elucidating mechanisms of lipid metabolism and microbiome regulation. scRNA-seq has greatly enriched our understanding of livestock research, and multi-omics integration enables a systematic understanding of the complex biology of livestock, supporting precision breeding and disease prevention.

## 5. Conclusions and Outlook

The rise of scRNA-seq has revolutionized livestock biology, offering insights into cellular heterogeneity at an unprecedented level. scRNA-seq has unveiled complex gene regulatory mechanisms across tissues, organs, and reproductive processes in livestock and poultry. When integrated with other omics technologies—such as epigenomics, proteomics, and spatial transcriptomics—single-cell technologies can provide multidimensional insights into critical biological processes, including growth, development, immunity, and metabolism. These technological integrations not only enhance the precision of biological process analyses but also offer valuable insights into the molecular mechanisms driving desirable livestock traits and production performance.

Looking forward, single-cell technologies are poised to continue evolving, with deeper integration into multi-omics platforms for a more comprehensive understanding of livestock biology. Emerging nanotechnologies and microfluidics are expected to improve the sensitivity and resolution of single-cell analysis, while artificial intelligence and machine learning algorithms will expedite data processing and the extraction of biological insights. These advancements will significantly propel the use of single-cell and multi-omics analyses in molecular breeding, disease prevention, productivity improvement, and environmental adaptability research, providing novel solutions and tools. Furthermore, the cost of scRNA-seq is likely to decrease, promoting its wider application in livestock production. High-quality single-cell atlases across various species will accumulate over time, ultimately contributing to standardized databases for gene expression and regulation in livestock. This progress will drive animal science towards more efficient, cost-effective, and sustainable practices.

## Figures and Tables

**Figure 1 ijms-25-12940-f001:**
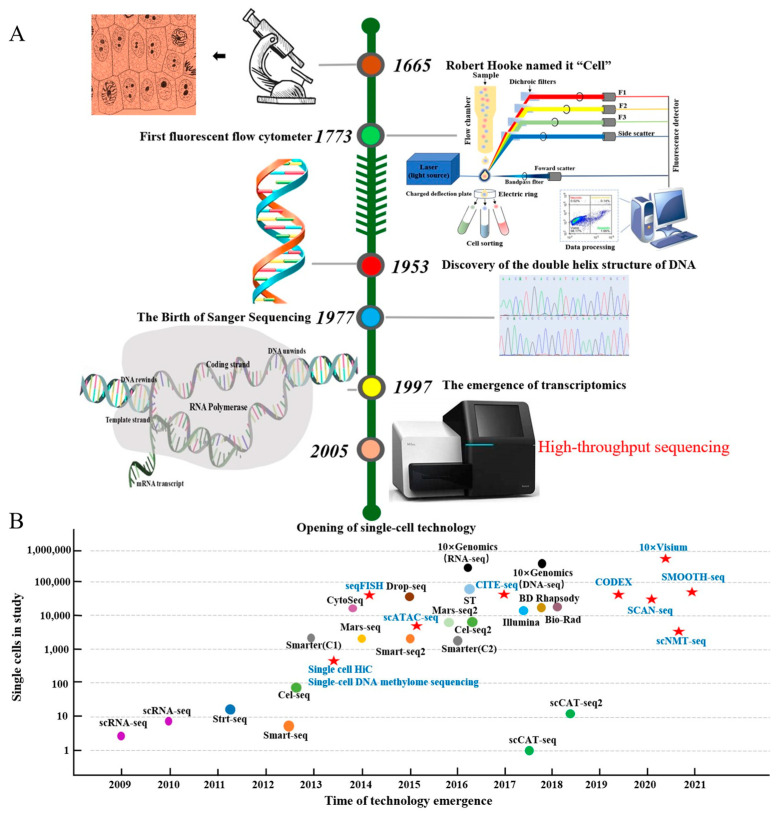
The discovery of cells and the evolutionary path of single-cell technology. Note: (**A**) shows the timeline from the discovery of the cell to the opening of the single-cell era, and (**B**) shows the single-cell methodology. The colored dots in (**B**) represent different single-cell techniques categorized by library construction or methodology, while red stars indicate approaches integrating single-cell and multi-omics technologies. Key methods include Strt-seq (single-cell tagged reverse transcription sequencing), Cel-seq (cell expression by linear amplification and sequencing), seqFISH (sequential fluorescence in situ hybridization), ST (spatial transcriptomics), CODEX (co-detection by indexing), Mars-seq (massively parallel single-cell RNA sequencing), scCAT-seq (single-cell chromatin accessibility and transcriptome sequencing), SCAN-seq (single-cell amplification and sequencing of full-length RNAs using nanopore technology), and scNMT-seq (single-cell nucleosome, methylation, and transcription sequencing).

**Figure 2 ijms-25-12940-f002:**
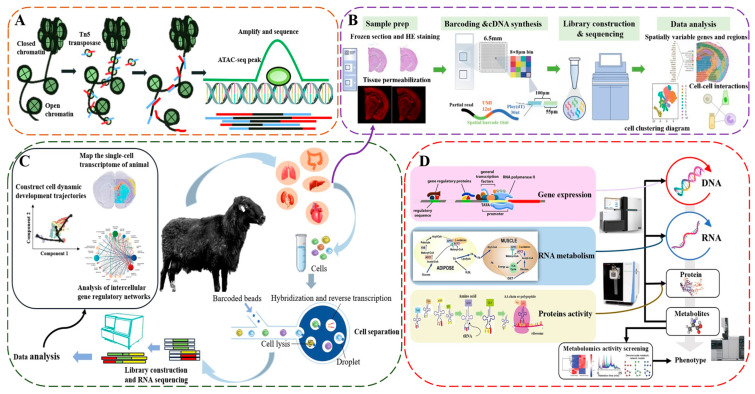
Principles or flowcharts of omics technology methods. Note: (**A**) ATAC-seq as an example of epigenetic technology, (**B**) 10× Genomics Visium as an example of spatial genomics technology, (**C**) 10× Genomics as an example of scRNA-seq technology, and (**D**) a conventional genomics technology.

**Figure 3 ijms-25-12940-f003:**
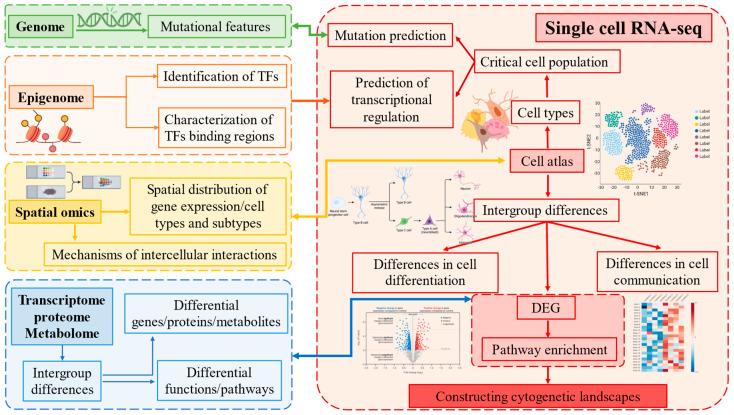
scRNA-seq and multi-omics technologies: cutting-edge tools for deciphering cell function, differentiation, and cross-cell communication.

**Table 1 ijms-25-12940-t001:** Characteristics of different single-cell isolation methods.

Methods	Number of Cells	Separation Mechanism	Speed	Advantages	Disadvantages
Serial Dilution	Large	Poor concentration control	Slow	Simple, cost-effective	Low purity, risk of contamination, inefficient for multi-cell isolation
FACS	Millions	Fluorescence detection, Light Scattering Measurement	Fast	High efficiency, accurate, multi-cell compatible	Expensive, high operational demand, some cell damage
Micromanipulation	Low	Cell visualization	Slow	High precision, direct observation	Time-intensive, complex, low throughput
Drop-Seq	Hundreds or thousands	Microfluidics	Fast	High throughput, suitable for multiple samples	Specialized equipment and technical support required
Immunomagnetic separation	Millions	Antibody-bound magnetic beads	Fast	High efficiency, high specificity	Potential for cell growth impact, risk of cell damage
LCM	Low	Visualization-based	Slow	High precision, maintains cell integrity	Expensive, complex, low throughput

## Data Availability

Not applicable.

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
