# Peer review of "Innovative Insights into Single-Cell Technologies and Multi-Omics Integration in Livestock and Poultry"

_ijms, 2024, doi:10.3390/ijms252312940_

Round 1

Reviewer 1 Report

Comments and Suggestions for Authors

Advances in Single-Cell Technologies and Multi-Omics Integration in Livestock and Poultry: Unveiling New Frontiers in Complex Biological Processes

Dear Authors,

The manuscript is very interesting, and well prepared. Describes very important issue involved with Single-cell technologies and Multi-Omics as important tools, which can helps improve animal health and productivity, meets with animal welfare aspect. Main problem in my opinion is lack of references in some parts of the text of manuscript and adaptation of references section to IJMS pattern described in Instructions for Authors.

Below I add some suggestions helpful in this process:

Lines 35-38

References must be added to this sentence.

Line 60

Maybe better to separate Figure 1 to A and B.

Lines 71-567

Line spacing can be adapted to this which was used in Introduction section.

Lines 87-93

References are required in this paragraph.

Lines 141-155

References required in first paragraph and more in second.

Lines 191-202

References required in first paragraph and more in first part second paragraph.

Lines 211-214

Reference is required in this sentence.

Lines 242-244

References are required in this sentence.

Lines 289-298

References are required in this paragraph.

Lines 303-306

Reference(s) is (are) required in this sentence.

Lines 361-365

References are required in this sentence.

Lines 409-414

Reference is required in this sentence.

Lines 496-500

References required in those two sentences.

Lines 526-528

References required in this sentence. Maybe reference 102 can be mentioned on the beginning of line 526: “…In studies Gu et al. [102] on metabolic disorders… (and without reference [102] in line 534).

Lines 578-816

References section

Must be adapted to the IJMS/MDPI patterns, available in Instructions for Authors (IJMS website).

Authors must be separated using semicolon in each reference.

Journal name abbreviations must be italicized.

Volume must be italicized.

Year of publication must be bold.

Ie. no. 2:

Lichtman, J.W.; Conchello, J.A. Fluorescence Microscopy. Nat. Methods 2005, 910-919. https://doi.org/10.1038/nmeth817

Line 579

Sci. Prog. as an abbreviation of Journal name must be added.

Please check also abbreviations in case of all references.

Author Response

Dear Reviewer,

We are grateful for the opportunity to revise our manuscript titled " Innovative Insights into Single-Cell Technologies and Multi-Omics Integration in Livestock and Poultry " (Manuscript ID: ijms-3316674). We appreciate the insightful comments and suggestions provided by the reviewers, which have significantly improved the quality of our work.

In response to the reviewers' feedback, we have made several key revisions to enhance the clarity, specificity, and impact of our manuscript:

The reviewer’s comment 1:

Lines 35-38 References must be added to this sentence.

Lines 87-93 References are required in this paragraph.

Lines 141-155 References required in first paragraph and more in second.

Lines 191-202 References required in first paragraph and more in first part second paragraph.

Lines 211-214 Reference is required in this sentence.

Lines 242-244 References are required in this sentence.

Lines 289-298 References are required in this paragraph.

Lines 303-306 Reference(s) is (are) required in this sentence.

Lines 361-365 References are required in this sentence.

Lines 409-414 Reference is required in this sentence.

Lines 496-500 References required in those two sentences.

Lines 526-528 References required in this sentence. Maybe reference 102 can be mentioned on the beginning of line 526: “…In studies Gu et al. [102] on metabolic disorders… (and without reference [102] in line 534).

Response 1:

Thanks to the valuable suggestions you have given, We have supplemented the missing parts of the literature to make this paper more complete.

The reviewer’s comment 2:

Line 60 Maybe better to separate Figure 1 to A and B. Lines 71-567 Line spacing can be adapted to this which was used in Introduction section.

Response 2:

Thanks for your suggestion. We mark A and B in Figure 1 and explain the specific details of A and B in the note section (Line 75-76). Lines 71-567 have had their line spacing adjusted.

The reviewer’s comment 3:

Lines 578-816 References section,Must be adapted to the IJMS/MDPI patterns, available in Instructions for Authors (IJMS website). Authors must be separated using semicolon in each reference. Journal name abbreviations must be italicized. Volume must be italicized. Year of publication must be bold. Ie. no. 2:

Lichtman, J.W.; Conchello, J.A. Fluorescence Microscopy. Nat. Methods 2005, 910-919. https://doi.org/10.1038/nmeth817

Line 579 Sci. Prog. as an abbreviation of Journal name must be added.

Response 3: Thank you very much for pointing out the issues and providing the template. The reference format has been fully revised and proofread according to the IJMS/MDPI model.

Yours sincerely,

Dr. Weidong Deng

Yunnan Provincial Key Laboratory of Animal Nutrition and Feed, Faculty of Animal Science and Technology, Yunnan Agricultural University, Kunming 650201, China

Tel.: +86-871-65220375

Email: dengwd@ynau.edu.cn

Reviewer 2 Report

Comments and Suggestions for Authors

ijms-3316674

This is a welcome summary on Single-Cell Technologies and Multi-Omics Integration data for Livestock and Poultry management. The authors after an accurate exploration of the literature provide detailed information on functional integration data on the Single Cell methodologies and related omics analysis. By the way form the same group of scientists just 2 years ago a similar paper was published please see here

Genes 202213(12), 2211; https://doi.org/10.3390/genes13122211

Therefore, although I found rigorous and correct the methodology of selection and evaluation of data here presented I think is better to modify the title in

«Advances in Single-Cell Technologies and Multi-Omics Integration in Livestock and Poultry: An updates of New Frontiers In Complex Biological Processes»

Instead

«Advances in Single-Cell Technologies and Multi-Omics Integration in Livestock and Poultry: Unveiling New Frontiers in Complex Biological Processes»

Author Response

Dear Reviewer,

We are grateful for the opportunity to revise our manuscript titled " Innovative Insights into Single-Cell Technologies and Multi-Omics Integration in Livestock and Poultry " (Manuscript ID: ijms-3316674). We appreciate the insightful comments and suggestions provided by the reviewers, which have significantly improved the quality of our work.

In response to the reviewers' feedback, we have made several key revisions to enhance the clarity, specificity, and impact of our manuscript:

The reviewer’s comment 1:

Form the same group of scientists just 2 years ago a similar paper was published please see here Genes 2022, 13(12), 2211; https://doi.org/10.3390/genes13122211.

Response 1:

We downloaded this article from genes and compared it to our upcoming paper at IJMS. In fact, although both articles, including the one from Genes published in 2022, originate from the same institution—College of Animal Science and Technology, Yunnan Agricultural University—there is no overlap in authorship between the two papers. Other points of similarity and difference are divided as follows:

Similarities

â‘ Research Focus: Both articles discuss single-cell technologies and multi-omics integration, highlighting their applications in livestock and poultry research with emphasis on cellular heterogeneity and gene regulatory networks.

â‘¡Coverage of Technologies: Both papers explore scRNA-seq and its integration with other omics layers, such as epigenomics, proteomics, and spatial transcriptomics.

â‘¢Application Scope: Both papers emphasize the use of single-cell technologies in poultry (e.g., chicken) and other economic animals (e.g., pigs, ruminants).

Differences

â‘ Research Focus: The Genes article focuses on the application of single-cell technologies in poultry, while the IJMS article covers a broader range of species, including poultry, pigs, and ruminants.

â‘¡ Depth of Multi-Omics Integration: The Genes article limits itself to single-omics data, whereas the IJMS article delves into the integration of single-cell data with other omics layers.

â‘¢ Data Analysis and Tools: The Genes article briefly mentions tools for single-cell data analysis, while the IJMS article provides a comprehensive review of various software and methods.

â‘£ Content Depth and Scope: The IJMS article offers a more comprehensive discussion of principles, methodologies, challenges, and future directions of single-cell technologies.

⑤ Publication Time and References: The Genes article was published in 2022 with 97 references, while the IJMS article was completed in October 2024, citing over 110 references, reflecting a broader literature base.

â‘¥ Figure Differences: The Genes article contains fewer and simpler figures. The IJMS article includes more diverse, visually appealing figures with advanced designs and vibrant color schemes.

The reviewer’s comment 2:

Therefore, although I found rigorous and correct the methodology of selection and evaluation of data here presented I think is better to modify the title in «Advances in Single-Cell Technologies and Multi-Omics Integration in Livestock and Poultry: An updates of New Frontiers In Complex Biological Processes», Instead «Advances in Single-Cell Technologies and Multi-Omics Integration in Livestock and Poultry: Unveiling New Frontiers in Complex Biological Processes»

Response 2:

Thanks for your suggestions. Some similarities were found but more differences, to avoid ambiguity, meanwhile we further modified the title to Innovative Insights into Single-Cell Technologies and Multi-Omics Integration in Livestock and Poultry.

Reviewer 3 Report

Comments and Suggestions for Authors

Comments about the manuscript:

“Advances in Single-Cell Technologies and Multi-Omics Integration in Livestock and Poultry: Unveiling New Frontiers in Complex Biological Processes”

Advances in single-cell RNA sequencing (scRNA-seq) methods have made it possible to advance our knowledge of regulatory networks and cellular dynamics, particularly in livestock and poultry. The review proposed for publication concerns the current state of applications of single-cell RNA sequencing in poultry, pigs and ruminants, in relation to reproduction and development. It concerns recent progress in the knowledge of single-cell integration with a look at its potential applications, particularly in the optimization of production performance and the prevention of diseases.

This work is useful. The manuscript is well constructed and well documented with a recent bibliography. It will be useful as a basis for researchers and as a source for teaching. I only have a few minor comments.

Line 61, figure 1. This table is called line 101. It would also be called in the introduction because it shows the evolution of the methods used

Line 101. Write “( Tab 1)” instead of “(see Tab 1)”

Line 543. I did not find the reference to table 3 in the text.

Author Response

Dear Reviewer,

We are grateful for the opportunity to revise our manuscript titled " Innovative Insights into Single-Cell Technologies and Multi-Omics Integration in Livestock and Poultry " (Manuscript ID: ijms-3316674). We appreciate the insightful comments and suggestions provided by the reviewers, which have significantly improved the quality of our work.

In response to the reviewers' feedback, we have made several key revisions to enhance the clarity, specificity, and impact of our manuscript:

The reviewer’s comment 1:

Line 61, figure 1. This table is called line 101. It would also be called in the introduction because it shows the evolution of the methods used

Response 1:

Thanks to your suggestion, we have added the labeling of Figure 1 at the end of the first paragraph of the introduction. We also separated Figure 1 into A and B to make the figure clearer.

The reviewer’s comment 2:

Line 101. Write “( Tab 1)” instead of “(see Tab 1)”

Response 2:

Thank you very much for your question, we have made changes.

The reviewer’s comment 3:

Line 543. I did not find the reference to table 3 in the text.

Response 3:

Thank you for your comment. The figures in the manuscript are based on a synthesis of data and findings from multiple studies referenced throughout the text. Since these figures are original summaries drawn from a range of sources, they do not correspond to a single specific reference. However, I have ensured that all the underlying studies and sources contributing to these figures are cited appropriately within the manuscript. In the article I marked Fig 3 at Line 574.

Reviewer 4 Report

Comments and Suggestions for Authors

The article highlights the potential of single-cell RNA sequencing (scRNA-seq) to analyse gene expression at the individual cell level and examines its revolutionary significance in cattle and poultry research. scRNA-seq offers comprehensive insights into cellular heterogeneity, exposing gene regulation systems essential for animal development, immunity, and reproduction, in contrast to bulk RNA-seq, which ignores cellular diversity. While scRNA-seq identifies genes linked to growth, muscle repair, and immunological activities in pigs, applications in poultry concentrate on characteristics like egg production and muscle growth. It improves precision breeding techniques in ruminants by tracking developmental stages in embryos and reproductive processes including spermatogenesis.

Additionally covered in the article is multi-omics integration, which combines scRNA-seq with techniques such as ATAC-seq, ChIP-seq, and spatial transcriptomics (ST). A three-dimensional picture of gene expression and cellular interactions may be obtained thanks to these combinations, which enable accurate mapping of gene regulation and cellular dynamics. By advancing knowledge of important pathways in livestock health, growth, and environmental adaptations, this integrated approach has laid the groundwork for better breeding, disease control, and production efficiency.

The article's title, "Unveiling the New Frontiers in Complex Biological Processes," accurately sums up its content. However, it is recommended to clarify some points and provide some corrections.

"The application of scRNA-seq has rapidly expanded..." is the first line that introduces research on poultry and animals. A more seamless transfer from the broad usage of scRNA-seq in biology to the specialized uses in poultry and livestock is needed, nevertheless.

Though a little ambiguous, the statement "single-cell resolution to unravel gene expression" makes clear that scRNA-seq offers information on the patterns of expression of specific genes at the single-cell level.

"Moreover, gene expression data captured by this technique is often limited to the RNA level...": Moving past technical constraints to integrating with other omics makes sense, but it would be helpful to include additional reasoning for the reason this integration is crucial for furthering research.

"The 10× Genomics system is one of the most widely used" : Although this statement is true, the wording may make it clearer that this platform specializes in high-throughput single-cell transcriptomics using microfluidic methods.

Without any background information or explanation, the phrase "scRNA-seq with Smart-seq2" is utilized. One particular scRNA-seq technique that needs to be described as a method is smart-seq2.

"A study applying scRNA-seq analyzed cell populations..." : It can be confusing to switch between studies. It should be noted that these investigations concentrate on different facets of pig biology. This may be made clearer by include a statement stating that each study focusses on a distinct biological system (immune, muscular, etc.).

The phrase: "Studies on Guanzhong dairy goats revealed that differentially expressed genes during spermatogenesis were primarily enriched in the Notch, Hippo, TGF-β, and stem cell pluripotency pathways" may be clarified. Clarifying what "differentially expressed genes" mean (for example, which genes or clusters) or if they pertain to upregulated or downregulated genes would be helpful.

The function of these genes in relation to developmental stages should be clarified in sections such as "uncovering temporally specific expression of key genes such as NOTCH1, TBX3, and FGFR4". It is insufficient to describe them as "key genes" without providing sufficient biological context.

There is ambiguity in 4.2. "For example, a study of spermatogenesis in sheep using scRNA-seq identified early and late primary spermatocytes, round spermatids, and other germ cells." There is no further clarification in the text as to how this observation relates to or contrasts with other discoveries. If it makes clear whether the study produced new understandings of these cell kinds or if it is merely a descriptive example, it could be more useful. The analogy to the Guanzhong pig study might be easier to understand with a more coherent explanation.

Minor revision

The terms "scRNA-seq" and "single-cell RNA-seq" were used interchangeably in the text. Selecting a single term and using it consistently throughout the article might make it more understandable.

Both "scRNA-seq emerged" and "scRNA-seq has rapidly expanded" characterize the technique's rise to prominence. Although both arguments have excellence, they essentially reiterate the same concept in slightly different ways without providing any additional details.

The term "bulk RNA-seq" is used to differentiate scRNA-seq from conventional RNA sequencing techniques. But the writing should be more explicit when stating that scRNA-seq entails sequencing individual cells, while bulk RNA-seq entails sequencing the transcriptomes of entire tissue samples.

"the rapid development of high-throughput multi-omics technologies in recent years" : This sentence from the introduction should be made simpler. Redundancy may result from its repetition later in the text in a slightly different way.

The absence of a space between "sorting" and the parenthesis in "fluorescence activated cell sorting(FACS)" could be a minor typo.

"Laser capture microdissection (LCM)": When using abbreviations in this way, it's ideal to make sure that each one is described in detail at the outset (for instance, "laser capture microdissection (LCM)" followed by "LCM").

"Providing a deeper understanding" seems more like a general conclusion than a specific finding in the sentence "Further, the primitive endoderm and epiblast in cattle blastocysts showed that TDGF1 and PRDM14 regulate early cell fate decisions, providing a deeper understanding of early embryonic cell fate determination." Maybe a more detailed explanation of the outcome would be preferable.

For improved flow, there are a few places where ideas or phrases are repeated and might be simplified. For instance, the idea that bulk RNA-seq and scRNA-seq are "complementary" or have different strengths is stated several times in several parts (e.g., 4.2, 4.3, 4.4). This information should be combined into a single, compelling explanation rather than being reiterated.

In 4.3, the phrase "This integrative approach provides deeper insights into the relationship between mRNA and protein levels, enabling a more precise characterization of cell types, cellular states, and dynamic changes under various developmental conditions." It could be simpler to say: "This integrative approach provides deeper insights into the relationship between mRNA and protein levels, which enables a more precise characterization of cell types, cellular states, and dynamic changes under various developmental conditions."

"Spatial transcriptomics (ST) was named the Method of the Year by Nature Methods in 2020, while scRNA-seq was named the Breakthrough of the Year." This could be rephrased as follows: "Spatial transcriptomics (ST) was also named Method of the Year by Nature Methods in 2020, even though scRNA-seq was named Breakthrough of the Year."

Comments on the Quality of English Language

The English language is well.

Author Response

Dear Reviewer,

We are grateful for the opportunity to revise our manuscript titled " Innovative Insights into Single-Cell Technologies and Multi-Omics Integration in Livestock and Poultry " (Manuscript ID: ijms-3316674). We appreciate the insightful comments and suggestions provided by the reviewers, which have significantly improved the quality of our work.

In response to the reviewers' feedback, we have made several key revisions to enhance the clarity, specificity, and impact of our manuscript:

The reviewer’s comment 1:

"The application of scRNA-seq has rapidly expanded..." is the first line that introduces research on poultry and animals. A more seamless transfer from the broad usage of scRNA-seq in biology to the specialized uses in poultry and livestock is needed, nevertheless.

Response 1:

Thank you for your valuable question and we have changed the text to read “scRNA-seq provides new tools for studying complex regulatory mechanisms in cell development, immune responses and reproduction. In poultry and livestock research, single-cell transcriptomics technologies have shown great potential, especially in profiling the immune system, developmental processes, and disease responses. For example, scRNA-seq has helped to reveal the specific roles of different cell types in the immune response or to reveal the transcriptomic profile of key cells during reproduction. However, single-cell analysis in livestock and poultry research still faces challenges such as cost, technical complexity, and data integrity, which limit its dissemination in practical applications.”

The reviewer’s comment 2:

Though a little ambiguous, the statement "single-cell resolution to unravel gene expression" makes clear that scRNA-seq offers information on the patterns of expression of specific genes at the single-cell level.

Response 2:

Thanks for the suggestions. In the text, amend 4 lines to read “To address this limitation, scRNA-seq emerged, providing single-cell resolution to reveal information about gene expression patterns, enabling a deeper understanding of cellular heterogeneity and regulatory biological processes (Tab 1).” More explicit expression of single-cell sequencing.

The reviewer’s comment 3:

"Moreover, gene expression data captured by this technique is often limited to the RNA level...": Moving past technical constraints to integrating with other omics makes sense, but it would be helpful to include additional reasoning for the reason this integration is crucial for furthering research.

Response 3:

Thank you very much for your suggestion. Chapter 4 of the text, “Application of Multi-Omics Integration with scRNA-seq in Livestock and Poultry”, we not only present the relevant applications of single-cell combined multi-omics, but also summarize the advantages of integrating the data.

The reviewer’s comment 4:

"The 10×Genomics system is one of the most widely used" : Although this statement is true, the wording may make it clearer that this platform specializes in high-throughput single-cell transcriptomics using microfluidic methods.

Response 4:

Changed the wording regarding 10×Genomics as per your suggestion. “Among scRNA-seq platforms, the 10×Genomics system is one of the most widely used. It is particularly good at high-throughput single-cell transcriptomics research using microfluidics.”

The reviewer’s comment 5:

Without any background information or explanation, the phrase "scRNA-seq with Smart-seq2" is utilized. One particular scRNA-seq technique that needs to be described as a method is smart-seq2.

Response 5:

10×Genomics and smart-seq2 are both methodological techniques for scRNA-seq, each of which has its own strengths, and the cited literature combines both techniques in order to capitalize on their respective strengths for a more comprehensive analysis of single-cell RNA expression. I have therefore changed the content to “In reproductive research, scRNA-seq based on 10×Genomics and Smart-seq2 identified distinct subpopulations of cumulus granulosa cells (cGCs) and mural granulosa cells (mGCs) within porcine antral follicles, revealing their differential roles in estrogen synthesis and hormone signaling.”

The reviewer’s comment 6:

"A study applying scRNA-seq analyzed cell populations..." : It can be confusing to switch between studies. It should be noted that these investigations concentrate on different facets of pig biology. This may be made clearer by include a statement stating that each study focusses on a distinct biological system (immune, muscular, etc.).

Response 6:

To provide a clearer understanding of the research aspects on which the literature focuses, we have changed the description to “A study applying scRNA-seq focused on skeletal muscle cell populations from newborn pigs of different breeds, identifying nine cell clusters.”

The reviewer’s comment 7:

The phrase: "Studies on Guanzhong dairy goats revealed that differentially expressed genes during spermatogenesis were primarily enriched in the Notch, Hippo, TGF-β, and stem cell pluripotency pathways" may be clarified. Clarifying what "differentially expressed genes" mean (for example, which genes or clusters) or if they pertain to upregulated or downregulated genes would be helpful.

Response 7:

We have added differential genes significantly enriched in Notch, Hippo, TGF-β and stem cell pluripotency pathways during spermatogenesis, such as EZH2, SCP2, PRKCD and PCNA.

The reviewer’s comment 8:

The function of these genes in relation to developmental stages should be clarified in sections such as "uncovering temporally specific expression of key genes such as NOTCH1, TBX3, and FGFR4". It is insufficient to describe them as "key genes" without providing sufficient biological context.

Response 8:

Thanks to your valuable suggestions, we have further refined the function of these key genes in relation to developmental outcomes in line 284 of the text, and add “Single-cell quantitative microfluidics was also used to identify early developmental genes in bovine embryos, and revealed that NOTCH1 is involved in early cell fate decision and tissue differentiation, and that the late stage-specific expression of TBX3 and FGFR4 is associated with cell proliferation and migration.”

The reviewer’s comment 9:

There is ambiguity in 4.2. "For example, a study of spermatogenesis in sheep using scRNA-seq identified early and late primary spermatocytes, round spermatids, and other germ cells." There is no further clarification in the text as to how this observation relates to or contrasts with other discoveries. If it makes clear whether the study produced new understandings of these cell kinds or if it is merely a descriptive example, it could be more useful. The analogy to the Guanzhong pig study might be easier to understand with a more coherent explanation.

Response 9:

According to the suggestion, we reintegrated the two references and found that the results obtained by the two different methods were slightly different, and in the Guanzhong pig study, the cell type identification results obtained by combining scRNA-seq and bulk RNA-seq data were more complete and convincing, and were modified as “ In contrast, combining these two approaches to study spermatogenesis in Guanzhong pigs resulted in the identification of not only spermatogonia, spermatogonia, and spermatocytes in the testis, but also three new types of somatic cells, in addition to the validation of germ cell types using bulk RNA-seq data, and the discovery of new key markers for porcine spermatogonial subpopulations CD99 and PODXL2”.

Minor revision 1:

The terms "scRNA-seq" and "single-cell RNA-seq" were used interchangeably in the text. Selecting a single term and using it consistently throughout the article might make it more understandable.

Response 1:

Modifications and harmonization have been made in response to the recommendations.

Minor revision 2:

Both "scRNA-seq emerged" and "scRNA-seq has rapidly expanded" characterize the technique's rise to prominence. Although both arguments have excellence, they essentially reiterate the same concept in slightly different ways without providing any additional details.

Response 2:

The phrases “scRNA-seq emerges” and “scRNA-seq rapidly develops” are the same concept, so we deleted them.

Minor revision 3:

The term "bulk RNA-seq" is used to differentiate scRNA-seq from conventional RNA sequencing techniques. But the writing should be more explicit when stating that scRNA-seq entails sequencing individual cells, while bulk RNA-seq entails sequencing the transcriptomes of entire tissue samples.

Response 3:

Thank you for your question, we have added in the introductory section, line 43, that bulk-RNA-seq is targeted at sequencing complete tissue samples.

Minor revision 4:

"the rapid development of high-throughput multi-omics technologies in recent years" : This sentence from the introduction should be made simpler. Redundancy may result from its repetition later in the text in a slightly different way.

Response 4:

We have changed line 37 of the text to “In recent years, high-throughput multi-omics technologies including genomics [7], epigenomics [8], transcriptomics [9], proteomics [10], and metabolomics [11] have been increasingly used in livestock and poultry research.” In order to reduce repetition.

Minor revision 5:

The absence of a space between "sorting" and the parenthesis in "fluorescence activated cell sorting(FACS)" could be a minor typo. "Laser capture microdissection (LCM)": When using abbreviations in this way, it's ideal to make sure that each one is described in detail at the outset (for instance, "laser capture microdissection (LCM)" followed by "LCM").

Response 5:

Thank you for your detailed suggestion, we have corrected it, where the full name of the LCM is in line 115.

Minor revision 6:

"Providing a deeper understanding" seems more like a general conclusion than a specific finding in the sentence "Further, the primitive endoderm and epiblast in cattle blastocysts showed that TDGF1 and PRDM14 regulate early cell fate decisions, providing a deeper understanding of early embryonic cell fate determination." Maybe a more detailed explanation of the outcome would be preferable.

Response 6:

Thanks to your suggestion, we have re-condensed the conclusions section of the paper, with the generalized conclusions changed to specific findings, as seen in line 290 of the text “Further, primitive endoderm and epiblast of bovine blastocysts showed that specific expression of TDGF1 and PRDM14 may be involved in early cell fate decisions.”

Minor revision 7:

For improved flow, there are a few places where ideas or phrases are repeated and might be simplified. For instance, the idea that bulk RNA-seq and scRNA-seq are "complementary" or have different strengths is stated several times in several parts (e.g., 4.2, 4.3, 4.4). This information should be combined into a single, compelling explanation rather than being reiterated.

Response 7:

To avoid redundancy, let's modify a few things

Line 426: “In conclusion, the use of bulk RNA-seq to acquire overall gene expression data, in conjunction with scRNA-seq that provides finer cell-level analysis, to collectively provide comprehensive insights that contribute to a deeper understanding of complex biological processes.”

Line 466 :“Integrating and analyzing scRNA-seq with proteomics data can provide a comprehensive view of cellular-level information to more accurately resolve complex biological processes and regulatory mechanisms.”

In addition, delete the duplicate in line 497 of the text.

Minor revision 8:

In 4.3, the phrase "This integrative approach provides deeper insights into the relationship between mRNA and protein levels, enabling a more precise characterization of cell types, cellular states, and dynamic changes under various developmental conditions." It could be simpler to say: "This integrative approach provides deeper insights into the relationship between mRNA and protein levels, which enables a more precise characterization of cell types, cellular states, and dynamic changes under various developmental conditions."

Response 8:

Thank you very much for giving you the content of the proposal, which has been modified see line 441. Modifications to make our articles more internationalized.

Minor revision 9:

"Spatial transcriptomics (ST) was named the Method of the Year by Nature Methods in 2020, while scRNA-seq was named the Breakthrough of the Year." This could be rephrased as follows: "Spatial transcriptomics (ST) was also named Method of the Year by Nature Methods in 2020, even though scRNA-seq was named Breakthrough of the Year."

Response 9:

Thank you very much for giving you the content of the proposal, which has been modified see line 482.
